# Effectiveness of COVID-19 shelter-in-place orders varied by state

Yevgeniy Feyman[1]*, Jacob Bor[2], Julia Raifman[1], Kevin N. Griffith[3]

**1** Department of Health Law, Policy, and Management, Boston University School of Public Health, Boston, MA, United States of America, **2** Department of Global Health and Department of Epidemiology, Boston University School of Public Health, Boston, MA, United States of America, **3** Department of Health Policy, Vanderbilt University Medical Center, Nashville, TN, United States of America

* yfeyman@bu.edu

**Data Availability Statement:** All relevant data are within the manuscript and its Supporting Information files.

## Abstract

State "shelter-in-place" (SIP) orders limited the spread of COVID-19 in the U.S. However, impacts may have varied by state, creating opportunities to learn from states where SIPs have been effective. Using a novel dataset of state-level SIP order enactment and county-level mobility data form Google, we use a stratified regression discontinuity study design to examine the effect of SIPs in all states that implemented them. We find that SIP orders reduced mobility nationally by 12 percentage points (95% CI: -13.1 to -10.9), however the effects varied substantially across states, from -35 percentage points to +11 percentage points. Larger reductions were observed in states with higher incomes, higher population density, lower Black resident share, and lower 2016 vote shares for Donald J. Trump. This suggests that optimal public policies during a pandemic will vary by state and there is unlikely to be a "one-size fits all" approach that works best.

## Introduction

During pandemics, public health interventions typically focus on either reducing population mobility or imposing physical barriers (such as mask mandates) to contain the spread of the pathogen [1]. Both sets of approaches are complementary. The effectiveness of such efforts depends partly on the public's compliance, which may vary with state-level factors. Between March 19th and April 7th, 2020, 39 states issued "shelter-in-place" (SIP) orders [2] to limit the spread of COVID-19. These orders direct residents to remain at home except for certain permitted activities. Nationally, SIPs are associated with reduced mobility, COVID-19 case incidence, and COVID-related mortality [3, 4]. However, local responses to SIPs varied markedly across states. Leveraging these natural experiments, we assessed the state-level impacts of SIPs on mobility, and whether these impacts were moderated by the state's demographic, political, or disease environments.

With the pandemic's future uncertain, information on where SIP orders were most effective by reducing population-level mobility is a critical input for policy makers, citizens, and healthcare providers alike. Understanding the effects of these policies, and how these effects varied is critical and timely. Thus, relying on data from Boston University on state-enacted policies and

**Funding:** The author(s) received no specific funding for this work.

**Competing interests:** The authors have declared that no competing interests exist.

mobility data from Google, we used a stratified regression discontinuity study design to examine the extent to which the effects of SIP orders was moderated by state-level factors.

## Materials and methods

### Data and population

This retrospective, county-level observational study leveraged data from a variety of sources. We first obtained daily changes in population mobility from Google Community Mobility Reports for February 15, 2020 through April 21, 2020. These reports contain aggregated, anonymized data from Android cellphone users who have turned on location tracking. Daily mobility statistics are presented for all U.S. counties, unless the data do not meet Google's quality and privacy thresholds [5]. We did not exclude any additional counties. Our analytic sample included 128,856 county-day observations, accounting for 1,184 counties and 125.3 million individuals.

We then identified dates of implementation for state SIP orders from the COVID-19 US State Policy (CUSP) database at Boston University [2]. Changes in state policies are identified and recorded from state government websites, news reports, and other publicly-available sources.

Lastly, we obtained a variety of county-level characteristics including daily data on COVID-19 deaths and cases from the Johns Hopkins Coronavirus Resource Center [6], results of the 2016 presidential election from the MIT Election Lab [7], and demographics from the Health Resources and Services Administration's Area Health Resource Files [8].

### Study variables

They key outcome in our analysis was changes in population-level mobility. Our data include the number of visits and time spent at various locations such as grocery stores or workplaces. Mobility is reported as the percent change relative to a median baseline value for the corresponding day of the week during Jan 3–Feb 6, 2020 [5]. We then computed an index measure by taking the mean of percent changes for all non-residential categories which included: retail and recreation, groceries and pharmacies, parks, transit stations, and workplaces.

Our key predictor of interest was the number of days, positive or negative, relative to SIP implementation. While SIP orders sometimes took effect at midnight (13), their effects on mobility may be delayed. In addition, when assessing other policies, there may be expectation of enactment. SIP orders, however, appear to have been announced with little to no delay between announcement and enactment [9]. Thus the announcement of these orders were associated with spikes in mobility as residents rushed to stock up on essential goods [10]. To account for this, and to capture actual response to the SIP order (rather than noise) we allowed for a one-day washout period. We thus dropped observations on the day of implementation.

We included additional variables in our analysis which may plausibly moderate the association between SIP orders and population mobility. First, the stage of the pandemic affects residents' exposure to COVID-19 and may change the effectiveness of policy efforts. Indeed, the WHO recommends different actions depending on pandemic phase [11]. We operationalized the stage of pandemic as measured by early vs. late enactment of a SIP (pre or post April 1st) and by the state-level cumulative number of COVID-19 deaths and cases on each SIP's effective date.

Second, population-level factors such as income have been found to be associated with a higher likelihood and ability to engage in social-distancing measures [12, 13]. In addition, structural racism [14] and disproportionate representation among "essential workers" [15, 16], may predispose Black individuals or individuals of Hispanic descent to additional risk. We

extracted the following county-level measures from the AHRF: population estimate (F11984-18); per-capita income (F09781-17); poverty rate (F13321-17); Black population (F13910-17); total area in square miles (F13874-10). We created a population-density measure by dividing population by total area in square miles, and a share of Black residents by dividing Black population by total population [8]

Third, recent research demonstrates that political beliefs affect individual-level responsiveness and perception of the pandemic and the likelihood of adoption of state policies [17–19]. We used county-level vote share for Donald J. Trump in the 2016 presidential election as a proxy measure for political partisanship. We believe this is an appropriate proxy of partisan political affiliation, as it captures not only political alignment but also potential agreement with Donald J. Trump's negative expressed views on lockdowns and SIP orders (for example, the President allegedly considered "reopening" as early as March) [20].

And lastly, most states saw dramatic declines in mobility before any SIP orders were enacted. It is possible that this so-called "private response" to the pandemic is related to the eventual enactment of SIP orders and/or their effectiveness. To operationalize this, we measured average state-level mobility change from February 15 to each state's date of SIP enactment.

All continuous variables were standardized and effect sizes reflect a one standard deviation change in the independent variable.

## Statistical analysis

We estimated both overall and state-specific local linear regression discontinuity (RD) models following the methods outlined by Calonico, Cattaneo, & Farrell (2020) [21]. We used the number of days relative to SIP orders as our running variable; e.g. a value of -1 indicates the day immediately preceding SIP implementation, and a value of 1 indicates the first day post-implementation. All estimates used robust bias-corrected standard errors clustered by county, with a triangle kernel and data-driven optimal bandwidth selection which is considered a best practice to reduce analyst subjectivity [21, 22]. This approach mimics randomization and allows us to observe the effects of SIP orders on mobility, since we would not expect counties to change day-to-day in other ways which would impact mobility. In sensitivity analyses, we also weighted our models by county population.

To investigate whether SIP order effects varied with state characteristics, we aggregated our county-level covariates to the state level by summing across counties within the state (cumulative deaths and cases) or taking either population-weighted averages (all other variables). To investigate whether our state-level RD estimates are statistically different from one another, we conduct a series of z-tests for each state-pair in our sample [23]. We then estimated a series of bivariable linear regressions with several state characteristics as our independent variables and our RD estimates for the effects of state SIP orders as the dependent variable, with robust standard errors.

First, to assess whether the population's prior exposure to the epidemic moderated SIP response, we compared state-level effect estimates by whether the order was early vs. late and by log-deaths and log-cases on each SIP's effective date. Second, to investigate whether SIP effect size was related to population risk factors for COVID transmission and adverse outcomes, we assessed whether SIP effect size varied with demographic and socioeconomic variables. Third, we assessed whether political alignment moderated the effect of SIP orders. Fourth, we assessed whether the pre-SIP decline in mobility was associated with the estimated effect of the SIP order.

Lastly, we also assessed whether SIP impact differed by whether the state had implemented a range of other measures at the time of the SIP order. SIP orders were implemented alongside complementary policies in some states, thus we investigated the possibility that other policies beyond SIP orders (enacted at the same time) may have had effects on mobility. To do so, we ran bivariable state-level models regressing the estimates RD coefficients on indicators for whether other policies (school closures, daycare closures, nursing home restrictions, business closures, restaurant closures, and gym closures) were enacted in the state. These indicators were obtained from the COVID-19 US State Policy (CUSP) database

Lastly, to understand how different categories of mobility (e.g., workplace, transit use etc.) responded to SIP orders, we re-ran our primary state-level RD specification with each mobility category as the outcome. We investigated variation in these coefficients across states, and investigated their association with the epidemiological and socioeconomic factors discussed earlier.

## Robustness checks

We conducted several robustness checks in our analysis. First, enactment of SIP orders was likely endogenous with respect to deaths and/or cumulative disease burden. Unlike standard linear or logistic regression approaches, effect estimates from regression discontinuity designs are not biased by covariates unless there are discontinuities in average covariate values at the threshold [22]. We investigate this by running our primary national RD specification with a 7-day lagged measure of deaths per million population as the outcome.

Second, while data-driven bandwidths tend to be superior to alternative bandwidth selection procedures [24], it is possible for variation in state-level results to arise due to variation in bandwidths. To account for this, we re-ran our RD model for each state with a fixed bandwidth taken from our primary model. We then examined whether the estimated coefficients are correlated with the estimated from our primary state-level RD specifications.

Third, to compare our estimates with existing work, we also implemented an event-study specification as an alternative to our primary RD model, following the approach in Allcott et al. which includes date and county fixed effects [25].

We considered a P-value of less than .05 as significant for RD models, and 0.10 as significant for the bivariate linear regressions due to small sample size (N = 39) [26]. Analyses were performed using Stata MP version 13.1 and the *rdrobust* package.

## Ethics statement

Our study involved secondary analysis of non-identifiable, aggregate data and therefore is not human-subjects research. No IRB approval was sought.

## Results

Mobility across states fell substantially prior to implementation of SIP orders. Across all counties, the average mobility decline prior to SIP enactment was 25.6 percentage points (95% CI: -28.3 to -22.9) but ranged from a decline of 48.7 to 3.2 percentage points. On average, SIP orders additionally reduced mobility by 12.0 percentage points (95% CI: -13.1 to -10.9; bandwidth +/- 2.9 days) (Fig 1). RD models weighted by county population produced similar estimates (-9.9, 95% CI:-13.2 to -6.7).

State-specific estimates varied widely: from -35 percentage points in Rhode Island (95% CI: -54.9 to -15.6; bandwidth +/- 3.6 days) to a 10.8 percentage point increase (95% CI: 7.6 to 14.0; bandwidth +/- 4.1 days) in Tennessee (Fig 2). Among all state-pairs, 58% of coefficients were statistically different from one another (S4 Table).

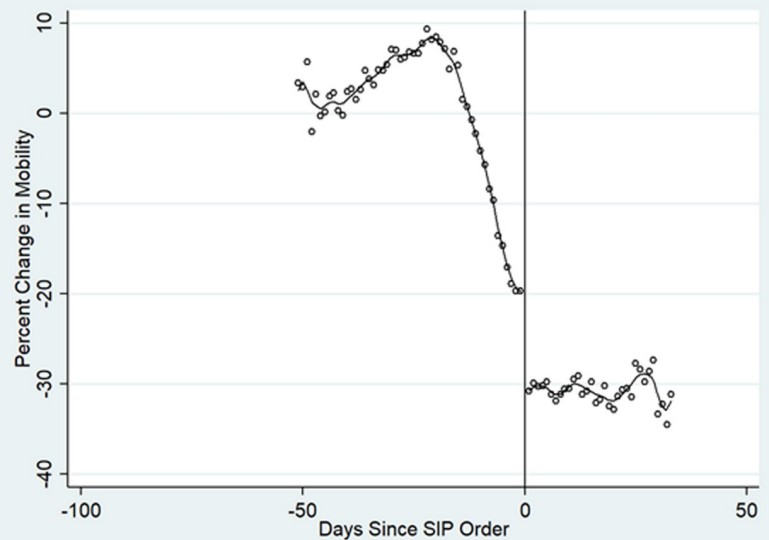

**Fig 1. Regression discontinuity plot for the relationship between SIP implementation and community mobility.**
Each dot is a binned daily average mobility relative to the baseline period. The trend-line is fit as a fourth order polynomial. Solid vertical line represents the relative date of SIP implementation.

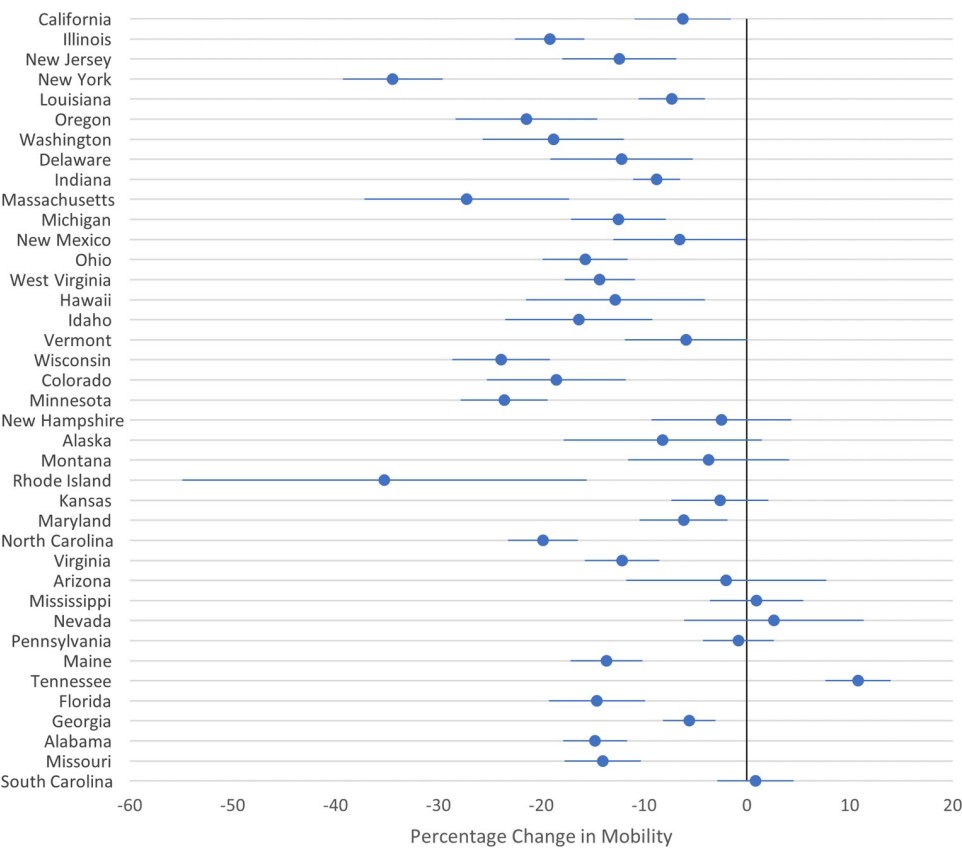

**Fig 2. State-specific regression discontinuity estimates.** Each dot is a coefficient estimate. Horizontal bars represent 95% confidence intervals.

We found larger declines in mobility following SIP orders in states with lower vote shares for Donald J. Trump in 2016 (-3.0% for a 12.3 percentage point change in vote share, 90% CI: 0.5 to 5.6), higher population densities (-3.4% per 1,000 people per square mile, 90% CI: -3.8 to -2.9), higher per capita incomes (-3.6% per $10,000, 90% CI: -5.4 to -1.8), and lower Black share of population (-1.8%, 90% CI: -0.1 to -3.5) (Table 1). Effect sizes did not differ significantly by state across early versus late adopters, poverty, mobility change prior to SIP enactment, and log-levels of COVID-19 deaths or cases at time of SIP.

In addition, we found some evidence that additional policies modified the effectiveness of SIP orders. School closures (-5.7%, 90% CI: -8.4 to -2.9) and non-essential business closures (-4.9%, 90% CI: -9.6 to -0.1) appear to be associated with more effective SIP orders, in terms of the effect on mobility. No other policies were significantly associated with variation in effectiveness of state-level SIP orders (Table 2).

We re-ran our primary RD specification using changes in mobility across each non-residential category separately and investigated the correlation of these measures. In general, all measures moved in the same direction as our index measure, and were strongly correlated with the index measure. The one exception was mobility at the workplace, which we hypothesize might be because essential workers continued to show up at work (S1 and S2 Tables).

As with our primary analysis, we find no relationship between cases or deaths for category-specific analyses. Consistent with the results for our index measure of mobility, Donald J. Trump vote share was associated with smaller reductions in mobility for parks (-20.8, 90% CI: -35.8 to -5.9) and transit (-3.45, 95% CI: -6.4 to -0.6). Our findings were similar for per capita income and poverty rate for certain categories as well. A unique finding from this analysis is that early SIP implementation appears to be associated with a large decline in mobility in the workplace category (7.54, 90% CI: 0.29 to 14.79) (S3 Table).

## Robustness checks

Below, we present the results of our robustness checks as discussed in the statistical analysis section.

## Disease burden

To test whether disease burden is a confounder in our RD models, we examined whether it has a discontinuity with the running variable. Our results (0.48, 95% CI: -0.04 to 0.99) indicate

**Table 1. Bivariable regression: Epidemiological factors.**

| | Coef. | 90% CI Lower | 90% CI Upper |
|---|---|---|---|
| **Donald Trump Vote Share (%)** | 3.0 (1.5) | 0.5 | 5.6 |
| **Population Density (1k per sqm)** | -3.4 (0.3) | -3.8 | -2.9 |
| **Per-Capita Income (in $10k)** | -3.6 (1.0) | -5.4 | -1.9 |
| **Black Share of Population (%)** | 1.8 (1.0) | 0.02 | 3.5 |
| **Early (vs. late) Adopters** | -5.5 (4.0) | -12.2 | 1.2 |
| **Poverty Rate (%)** | 2.0 (1.2) | 0.01 | 3.9 |
| **Mobility Change pre-SIP** | -0.8 (0.9) | -2.4 | 0.7 |
| **Natural Log of Deaths** | 2.2 (1.9) | -1.0 | 5.5 |
| **Natural Log of Cases** | 0.7 (4.1) | -6.3 | 7.6 |

Coefficient estimates for continuous variables represent the estimated effect of one standard deviation change in the predictor. A positive coefficient represents a smaller reduction in mobility. N = 39. Robust standard errors are in parentheses.

**Table 2. Bivariable regression: Other policies.**

|  | Coef. | 90% CI Lower | 90% CI Upper |
|---|---|---|---|
| **School Closure** | -5.7 (1.6) | -8.4 | -2.9 |
| **Daycare Closure** | -5.2 (3.2) | -10.5 | 0.2 |
| **Nursing Home Restrictions** | -0.6 (3.0) | -5.6 | 4.4 |
| **Non-essential Business Closure** | -4.9 (2.8) | -9.6 | -0.1 |
| **Restaurant Closure** | -4.5 (3.3) | -10.0 | 1.0 |
| **Gym Closure** | -3.2 (3.0) | -8.2 | 1.8 |

A positive coefficient represents a smaller reduction in mobility. N = 39. Robust standard errors are in parentheses.

that while it is plausible that SIP enactment is endogenous to disease burden, it is nonetheless not a confounder for our analysis.

## Variable bandwidth

While estimates from state-level models with a fixed bandwidth differed from estimates that allowed the bandwidth to vary by state, the estimated coefficients were strongly correlated (r = 0.77) (S2 Fig).

## Event study design

Replicating prior work that uses an event study design [25] we find that the estimate for one day after SIP implementation (-7.9, 95% CI: -8.5 to -7.5) is similar to our primary RD result (S3 Fig). We note that whereas the event study compares means on either side of the threshold, the RD approach leverages data points away from the threshold in order to obtain more precise estimates of state-specific effects.

## Discussion

States experienced significant heterogeneity in mobility following SIP implementation. Whereas SIPs led to large reductions in population mobility for New York, Rhode Island, and Wisconsin; they did not significantly reduce mobility in Pennsylvania, Kansas, or Mississippi. Mobility increased in Tennessee during the days immediately following that state's SIP order. This unexpected finding could be due to the fact half of the state was already under a local shelter-in-place order, including Tennessee's metropolitan areas [27].

We observed large mobility reductions in the days preceding a SIP order. This suggests if individuals do not feel safe from COVID-19 in their communities, private responses will sharply reduce mobility even in the absence of a SIP order. Perhaps surprisingly, the magnitude of SIP effects was not associated with the date of the SIP order and the number of cases and deaths at the time of the SIP order. Notably, effect sizes were not correlated with changes in mobility preceding the SIP order. This suggests that responses to government policies and private responses to COVID were not substitutes. More explicitly, it implies that mobility changes after the SIP order were not simply a function of a pre-existing trend in the state.

Less surprisingly, we found that certain policies–school closures and non-essential business closures–were complementary to SIP orders. It may be that these policies reduce burdens that would otherwise limit the ability of individuals to comply with SIP orders.

While the variation in the effectiveness of SIP orders in reducing mobility is likely to partly be a function of other policies implemented concurrently, this is unlikely to be the driver of the effect heterogeneity that we observe. Indeed, nearly all states in our sample (38 and 32,

respectively) implemented the policies that have a relationship with our estimated RD coefficients (school closures and business closures). Moreover, even among states with every policy implemented, there remains substantial variation in effect size (mean: -14.8, SD:11.5) Future research should explore the potential mechanisms underlying these differences in more detail.

Mobility responses to SIPs were also associated with political and social factors independent of local epidemiological conditions. Greater changes in mobility in response to SIPs were observed in states with lower shares of Trump voters, higher population density, higher per capita income, and lower share of Black residents (which may be partly be a proxy for the concentration of essential workers) [15]. This was generally consistent with our analysis of category-specific changes in mobility. Our findings are consistent with evidence finding differential individual pandemic responses due to political beliefs [17] and in high-density areas whether the expected benefits of distancing are greater.

However, in examining category-specific mobility changes, we *do* find that earlier SIP implementation appears to be associated with larger declines in visits to workplaces. While we cannot identify the precise mechanism for this, it may be that some private responses may have been counter to the public response (e.g., employers attempting to stay open).

In addition, while not directly comparable, our analysis is consistent with recent work finding that income is associated with the ability to comply with social distancing orders [13, 28]. Our results are also similar in magnitude to other work using event-study designs [25], and we come to similar conclusions that a large component of the overall mobility reduction occurred due to a private rather than policy-driven response.

Notably, a strength of our approach is that while policy enactment might be endogenous to pre-policy conditions, our estimates are identified entirely from the discontinuous change that occurs post-enactment.

## Limitations

There are limitations to our study. First, we focused on the short-run, immediate impact of SIPs on mobility. While SIPs may have had longer-run impacts too, we could not separate these effects from other secular trends in our RD study design. Second, it is unclear whether the Android users who contribute data to Google Community Mobility Reports are representative of individuals living in that county. Unfortunately, this information is unavailable. Indeed, there are observed demographic differences between Android and iPhone users across age groups and gender [29]. To the extent they are different, our results may not generalize to other individuals. Third, other policies (such as transportation or business closures) were implemented either in tandem with or prior to SIP order enactment. These policies likely had their own effects, and may have affected compliance with SIP orders, which may explain some of the variation between states. Fourth, while some other work has relied on county-level SIP orders as well, our analysis has not accounted for these changes. Any pre-existing county-level SIP orders may attenuate our effect estimates. Thus, our results may be a conservative estimate. Lastly, small sample sizes in our bivariable regressions may limit our ability to detect associations with state-level effect estimates. Despite this limitation, we are able to identify associations between our effect estimates and certain important measures such as political affiliation and some demographics. Moreover, the heterogeneity that we find is consistent with prior evidence on heterogeneity in mobility and responsiveness to SIP orders."

## Conclusion

Our results suggest that policies to deal with COVID-19 as well as future pandemics will have to account for population-level behavioral responses to those policies. Indeed, not everyone

can or will be able to comply with SIP policies. This is shaped by myriad factors including structural racism, and it likely means that complementary policies (e.g., mask mandates) may be essential.

Future research should use our results to elucidate mechanisms for why SIP orders worked in some states and not others, and why certain states were able to see large declines in mobility despite their group averages.

## Supporting information

**S1 Fig. State-level RD estimates.** State-specific RD plots using the main analytic approach described in the manuscript.
(TIF)

**S2 Fig. Scatter plot of state-level RD estimates with and without a fixed bandwidth.** These estimates are generated using the approach described in the manuscript. The variable bandwidth allows the bandwidth to vary by state. The fixed bandwidth uses a single bandwidth of 2.9 for all states. This bandwidth is taken from the primary national specification. The correlation between the two is 0.77.
(TIF)

**S3 Fig. Event-study specification.** These estimates are from an event-study specification, with the x-axis reflecting days relative to SIP order enactment. The specification includes date and county fixed effects. Leads and lags are censored at +/- 21, and estimates/CIs are accumulated to that point. The outcome is mean mobility, and the coefficient of interest is on the day after SIP enactment.
(TIF)

**S1 Table. RD estimate by mobility category.** This presents mean state-level coefficients and confidence intervals for each mobility category. RD methods are as described in the manuscript.
(DOCX)

**S2 Table. Correlation matrix of category-specific RD estimates.** This is a state-level correlation matrix of RD estimates of the effect of SIP orders on each category of mobility. There is low to high correlation across all estimates. Changes in workplace mobility appears to be negatively correlated with changes in mobility for all other categories.
(DOCX)

**S3 Table. Mobility category-specific bivariable results.** These are state-level, category-specific bivariable results regressing coefficients on the socioeconomic and epidemiological factors. Coefficient estimates for continuous variables represent the estimated effect of one standard deviation change in the predictor. A positive coefficient represents a smaller reduction in mobility. N = 39. Robust standard errors are used.
(DOCX)

**S4 Table. Z-statistics for difference in state-level coefficients.** This table reports z-statistics testing whether each state-pair in our sample has statistically different coefficients. Out of 741 tests, 429 (58%) are statistically different from one another. Green shading indicates a z-statistic greater than 1.96.
(DOCX)

**S1 File. Full set of data and code for replication.**
(ZIP)

## Author Contributions

**Conceptualization:** Yevgeniy Feyman, Jacob Bor, Kevin N. Griffith.

**Formal analysis:** Yevgeniy Feyman, Kevin N. Griffith.

**Investigation:** Yevgeniy Feyman.

**Methodology:** Yevgeniy Feyman, Jacob Bor, Julia Raifman, Kevin N. Griffith.

**Supervision:** Jacob Bor, Julia Raifman, Kevin N. Griffith.

**Validation:** Kevin N. Griffith.

**Writing – original draft:** Yevgeniy Feyman, Kevin N. Griffith.

**Writing – review & editing:** Yevgeniy Feyman, Jacob Bor, Julia Raifman, Kevin N. Griffith.

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
