## [Decision Letter · Decision Letter 0]

28 Oct 2020

PONE-D-20-29575

Effectiveness of COVID-19 Shelter-in-Place Orders Varied By State

PLOS ONE

Dear Dr. Feyman,

Thank you for submitting your manuscript to PLOS ONE. After careful consideration, we feel that it might have merit but does not fully meet PLOS ONE’s publication criteria as it currently stands. Therefore, we invite you to submit a revised version of the manuscript that addresses the points raised during the review process.

Please be thoughtful in your revisions. Given some of the comments, it is possible you may decide that you need to make major changes with different datasets and different analyses. 

We look forward to receiving your revised manuscript.

Kind regards,

Jaymie Meliker, Ph.D.

Academic Editor

PLOS ONE

Journal Requirements:

2. In the Methods section of the manuscript, please address the following:

-    Please provide a justification for why Donald J Trump was selected as the political proxy.

-    Please provide a citation for the AHRF database, the date ranges over which you conducted the search, which variables were studies. And ensure that you have provided sufficient details that others could replicate the analyses

Reviewers' comments:

Reviewer's Responses to Questions

**Comments to the Author**

1. Is the manuscript technically sound, and do the data support the conclusions?

Reviewer #1: Partly

Reviewer #2: Partly

Reviewer #3: Yes

Reviewer #4: No

2. Has the statistical analysis been performed appropriately and rigorously? 

Reviewer #1: Yes

Reviewer #2: Yes

Reviewer #3: Yes

Reviewer #4: No

3. Have the authors made all data underlying the findings in their manuscript fully available?

Reviewer #1: Yes

Reviewer #2: Yes

Reviewer #3: Yes

Reviewer #4: Yes

4. Is the manuscript presented in an intelligible fashion and written in standard English?

Reviewer #1: Yes

Reviewer #2: Yes

Reviewer #3: Yes

Reviewer #4: Yes

5. Review Comments to the Author

Reviewer #1: See attached.

I find the manuscript relevant, clear, and technically sound, for the most part. I would endorse its publication if the authors successfully address the points raised below.

The biggest threat, as I see it, to the analysis’s first two findings, with implications for the third, is that other time-varying COVID-relevant policies were simultaneously taking place, potentially confounding the analysis.

Reviewer #2: The authors examine the heterogeneous effects of SIPs on mobility patterns. They use an RD design to estimate state-specific effects of SIP on mobility patterns and the correlate the estimated effects with state covariates. Overall, the paper is succinct, well-written, and on an important topic. My concerns are listed below.

Other drivers of heterogeneity. There are several aspects of the analysis that may cause heterogeneous estimates even if the effect of the policies are the same.

- The RD bandwidth differs across states. Moreover, the "private" or voluntary distancing behaviors may differ across these bandwidths. It would be good to (a) examine robustness to a fixed bandwidth for the RD, and (b) examine robustness to using an event-study specification as in Allcott et al. (2020; "Economic and Health Impacts of Social Distancing Policies during the Coronavirus Pandemic"). E.g., What is the correlation between the current state-level estimates and the state-level estimates from (a) and (b)?

- The authors take the mean across categories in the Google Mobility Reports. Different states may have different shares of categories. It would be good to also examine each category individually as robustness to make sure the results agree. Again, what is the correlation between the state-level estimates using the mean index and the state-level estimates using each individual category?

- The authors do not account for county-level SIPs. Different states may have different shares of county SIPs already---driving observed heterogeneity. See Allcott et al. (2020; "Economic and Health Impacts of Social Distancing Policies during the Coronavirus Pandemic") for treatment of county level SIPs.

Correlates of estimated treatment effects.

- Small sample size is not a good reason to change the standard of statistical significance. The authors should just report the coefficient estimates along with the standard errors. The reader can then easily perform any required statistical tests.

- It would be useful to see multivariate analysis of all covariates in addition to the bivariate analysis that is already conducted.

The authors should really have standard RD plots for each state estimate in the appendix so the reader can visualize the extent to which they think the RD assumptions are valid.

Literature:

- Painter and Qiu (2020; "Political beliefs affect compliance with COVID-19 Social Distancing Orders") should be cited because it is most relevant to examining heterogenous effects in response to SIPs by partisanship.

- Please also go through https://scholar.google.com/scholar?cites=939500896822965987 to make sure all relevant literature is being cited for heterogenous effects of SIPs (e.g., Wright et al.).

- The findings in this manuscript should then be compared to what others have found.

Reviewer #3: The paper studies the effectiveness of COVID-19 Shelter-in-Place Orders among states. A novel dataset is constructed and analyzed by a state-of-the-art regression discontinuous method.

Overall I think this paper is complete and clearly written. The findings are interesting. I do have some reservations which I hope the authors can address. My detailed comments and questions are given below.

1. In Line 44, "SIP orders reduced mobility nationally by 12 percentage points."

2. The observations are Android cellphone users who have turned on location tracking. Are these observations representative of the population? Could the authors provide some summary statistics of demographics of such a sub-population?

3. In Line 120, it is unclear how the index measure is constructed. The index measure is the mean of what?

4. In Line 129, the observations on the day of implementation are dropped. I wonder how the estimation results could change without dropping those observations. The SIP orders should be expected, so I am not sure if the washout period should be taken into account. The authors may give more justication or reference.

5. In Figure 2, the label of the horizontal axis is \\Days from SIP Implementation." Shouldn't it be \\percentage change in mobility"? I don't see how the relative date of SIP implementation reflects in the figure. Is California the last to implement SIP, and is South Carolina the first?

Reviewer #4: This paper uses a regression discontinuity design to estimate the effect of shelter-in-place (SIP) orders on mobility using county-level mobility data from Android cellphone users.

Major comments:

* An important factor that may explain the findings are differences in how the SIP and related measures were implemented in each state. For example, some states had more enforcement, more stringent measures such as restrictions on religious and other public events, maximum number of people allowed in social gatherings, closing of different types of businesses (indoor and outdoor dining, bars, gyms, stores), etc. Policies aiming at restricting mobility also differed widely at the county and city level in each state. Hence, part of the difference in effects may be due to differences in how the SIP and accompanying policies were implemented across states. The estimated differences across states may therefore represent responses to different policies (or bundles of policies), not differential responses to a homogeneous policy implemented uniformly across states.

* The differences between the estimates of mobility changes across states do not seem statistically significant in most cases (Figure 2). Hence much of this estimated variation may be due to sampling variability, not necessarily to real heterogeneity in the effect of the SIP orders.

* The regressions shown in Table 1 use only 39 observations, which seems very low. Lack of significance in the results for epidemiological factors may be due to lack of statistical power and not a true lack of effect. The statistical analysis does not convincingly justify the conclusions on page 12 (lines 265-273).

Minor comments:

* The data only include Android cellphone users. The authors should discuss whether they think this subgroup is representative of each state's population.

* I was confused by the second paragraph on page 8 and the note under Table 1 stating that standard errors are clustered at the state level. My understanding is that the data for these regressions is already at the state level, which means each cluster has a single observation (so standard errors are heteroskedasticity-robust but do not effectively allow for clustering in any useful way).

* It would be interesting to analyze changes in mobility after the SIP is lifted to see whether mobility reverts back to its previous levels or remains lower.

6. PLOS authors have the option to publish the peer review history of their article (what does this mean?). If published, this will include your full peer review and any attached files.

Reviewer #1: **Yes: **Jonathan Rothwell

Reviewer #2: No

Reviewer #3: No

Reviewer #4: No

---

## [Author Response · Author response to Decision Letter 0]

22 Nov 2020

We have attached the response letter and have replicated it below:

Comments from editor

We have updated all formatting to be consistent with these requirements.

 2. In the Methods section of the manuscript, please address the following:

- Please provide a justification for why Donald J Trump was selected as the political proxy.

We have added the following in the methods section:

“We believe this is an appropriate proxy of partisan political affiliation, as it captures not only political alignment but also potential agreement with Donald J. Trump’s negative expressed views on lockdowns and SIP orders (for example, the President allegedly considered “reopening” as early as March).”

- Please provide a citation for the AHRF database, the date ranges over which you conducted the search, which variables were studies. And ensure that you have provided sufficient details that others could replicate the analyses

We have added the reference to the AHRF and have added the following text:

“We extracted the following county-level measures from the AHRF: population estimate (F11984-18); per-capita income (F09781-17); poverty rate (F13321-17); Black population (F13910-17); and total area in square miles (F13874-10). We created a population-density measure by dividing population by total area in square miles, and a share of Black residents by dividing Black population by total population.”

We have added the following ethics statement:

“Our study involved secondary analysis of non-identifiable, aggregate data and therefore is not human subjects research. No IRB approval was sought.”

We have removed this and have added the numerical results in the text.:

“RD models weighted by county population produced similar estimates (-9.9, 95% CI:-13.2 to -6.7).”

We have added these captions, as well as additional supporting information files. We have also removed figures from the main manuscript and uploaded them separately.

Comments from Reviewer 1

I find the manuscript relevant, clear, and technically sound, for the most part. I would endorse its publication if the authors successfully address the points raised below.

The paper has three major findings: 1) the mean effect of shelter-in-place orders 2) the variance in those effects by state 3) a list of mediating variables.

The biggest threat, as I see it, to the analysis’s first two findings, with implications for the third, is that other time-varying COVID-relevant policies were simultaneously taking place, potentially confounding the analysis.

These include: school closures, closure/restrictions on public transportation, restrictions on gatherings, and the closures of non-essential businesses, to name just three types of policies that varied by state on a day-to-day basis during the time period of analysis. Each of these could affect mobility data.

At the very least, the authors need to devote some space to summarizing analysis and discussion of these policies and why they don’t need to be explicitly modeled alongside SIP orders. My sense is that SIP orders left non-essentially businesses open in some states (particularly Democratic Party-oriented states) but not others.

So, one thing the authors need to consider is that not all SIP orders took the same form, and that might explain some of the variation in efficacy on mobility.

We thank the reviewer for taking the time to review our manuscript and suggesting ways to improve upon our work. We agree that each state has a unique policy environment which may moderate/mediate the effectiveness of state SIP orders. While a review of these myriad policies is outside the scope of this text, we have clarified in the text that: (a) SIP orders may have occurred alongside other restrictions implemented at the same time and our results should be interpreted as reflecting the aggregate impact of all policies implemented simultaneously with SIPs (b) implementation of other policies at the same time (e.g. closures of transportation, non-essential businesses, etc.) could have encouraged and enabled greater compliance with the SIPs and could be one factor explaining differences across states. In our revision, we ran the following analysis: using the BU database of policy responses, we obtained information on whether there were school closures, daycare closures, nursing home restrictions, business closures, restaurant closures, and gym closures. We then ran several regressions of our estimates of SIP effectiveness on indicators for these policies. Only school and non-essential business closures were associated with variation in the RD estimate, both being associated with more effective SIP orders. Notably, only one state did not enact a school closure in our time period (California). We have added the following paragraph in the results section:

“We found some evidence that additional policies modified the effectiveness of SIP orders. School closures (-5.7%, 90% CI: -8.4 to -2.9) and non-essential business closures (-4.9%, 90% CI: -9.6 to -0.1) appear to be associated with more effective SIP orders. No other policies were significantly associated with variation in effectiveness of state-level SIP orders. (Table 2)” 

We have added the following to the limitations section:

“Third, other policies (such as transportation or business closures) were implemented either in tandem with or prior to SIP order enactment. These policies likely had their own effects, and may have affected compliance with SIP orders, which may explain some of the variation between states.”

There is one more serious threat to accurate identification: the disease burden. The Northeast (especially New York City) was hit harder and earlier than much of the rest of the country. I don’t see anything in the authors’ discussion or code that indicates that the disease burden (ideally measured as deaths per capita) was considered as a control variable alongside the SIP order, but surely, an explosion of deaths and infections in a local area or state would be a relevant consideration as to whether or not people feel comfortable venturing out—and whether or not a state governor or legislative body issues a SIP order in the first place.

The reviewer highlights an important point. Unlike standard linear or logistic regression approaches, effect estimates regression discontinuity designs are not biased by covariates unless there are discontinuities in average covariate values at the threshold (Athey & Imbens, Journal of Economic Perspectives, 2017). We thus tested for these discontinuities in the revised manuscript:

Even without such a discontinuity, disease burden may be an important moderating factor for SIP effectiveness. In the revised version, we

Along these lines, there is a serious threat to endogeneity. The SIP orders were not issued randomly. Governors decided to issue them or not based on relevant political party alignment and the disease burden in their jurisdiction. I believe this issue needs to at least addressed under “limitations” and, potentially, the methods section, even if there is no analytic solution, other than robustness checks that include the disease burden in the main estimations. 

This is an important point. We agree with the reviewer that enactment of SIP orders is likely endogenous to disease conditions and political considerations. Although this is a problem in most analyses of SIPs, the regression discontinuity approach offers a more compelling and better identified way to get at policy impact (see response to previous comment). We identify solely off of the discontinuous change that occurs on the date of SIP implementation. Therefore, while pre-SIP conditions could spur the implementation of a SIP, the change in mobility immediately after a SIP is put into place would be unlikely to occur in the absence of the SIP. We have clarified that this is a strength of our analysis in the discussion section as follows:

“Notably, a strength of our approach is that while policy enactment might be endogenous to pre-policy conditions, our estimates are identified entirely from the discontinuous change that occurs post-enactment.

We have added the following to our results section:

“To test for whether disease burden is a confounder, we examined whether it has a discontinuity with the running variable. Our results (0.48, 95% CI:-0.04 to 0.99) indicate that while it is plausible that SIP enactment is endogenous to disease burden, it is nonetheless not a confounder for our analysis.”

Otherwise, I think the article is well structured, and I think the discussion and analysis of factors that distinguish effect sizes across states is valuable and consistent with other evidence on variation in compliance: https://papers.ssrn.com/sol3/papers.cfm?abstract_id=3638373

We thank the reviewer for these helpful comments and have added the reference in our revision. 

Comments from Reviewer 2

Other drivers of heterogeneity. There are several aspects of the analysis that may cause heterogeneous estimates even if the effect of the policies are the same.

- The RD bandwidth differs across states. Moreover, the "private" or voluntary distancing behaviors may differ across these bandwidths. It would be good to (a) examine robustness to a fixed bandwidth for the RD, and (b) examine robustness to using an event-study specification as in Allcott et al. (2020; "Economic and Health Impacts of Social Distancing Policies during the Coronavirus Pandemic"). E.g., What is the correlation between the current state-level estimates and the state-level estimates from (a) and (b)?

We agree that the use of different bandwidths across states may add variability to our estimates. We have re-run each state model using the fixed bandwidth of 2.9 that was used for the overall model. The results are presented below, and are highly correlated (r=0.77). We believe that allowing for variable data-driven bandwidths is more appropriate as this approach allows for more flexibility in estimating state-specific levels of mobility immediately before and immediately after SIP implementation. Prior work has illustrated that data-driven bandwidth selection tends to outperform alternative approaches. (Calonico, Cattaneo, and Titunik 2014) That is, use of variable bandwidths is consistent with the notion of identification at the threshold in an RDD design. 

We have added the following to the statistical analysis section:

“Second, while data-driven bandwidths tend to be superior to alternative bandwidth selection procedures, (21) it is possible for variation in state-level results to arise due to variation in bandwidths. To account for this, we re-ran our RD model for each state with a fixed bandwidth taken from our primary model. We then examined whether the estimated coefficients are correlated with the estimated from our primary state-level RD specifications.”

In addition, we have added the figure below to the appendix, and have added the following to our results section: 

“While estimates from state-level models with a fixed bandwidth differed from estimates that allowed the bandwidth to vary by state, the estimated coefficients were strongly correlated (r=0.77)”

We also thank the reviewer for pointing us to the recent work by Allcott et al. We do note that this is a fundamentally different identification strategy and requires different assumptions than our RD approach. We conducted two analyses: 

First, we re-ran our RD specification with a uniform kernel and ran an event study specification without any fixed effects to estimate both models in as similar a manner as possible. In the RD approach the coefficient estimate is -11.1 (SE:0.38) and in the event study approach the coefficient estimate is equal (-11.1, SE: 0.38).

The event study graph is presented below.

Next, we replicated the event-study specification with the same restrictions (up to 21 leads/lags) and date and county fixed effects as used by Allcott et al. Here, the coefficient is smaller (-7.9, SE: 0.24) than our primary RD estimate (-12.03, SE: 0.84) but is similar in magnitude. We’ve presented the overall result below and have added it to the appendix. 

We have also added the following to our results section:

“Replicating prior work that uses an event study design (15) we find that the estimate for one day after SIP implementation (-7.9, 95% CI: -8.5 to -7.5) is similar to our primary RD result. We note that whereas the event study compares means on either side of the threshold, the RD approach leverages data points away from the threshold in order to obtain more precise estimates of state-specific effects.”

- The authors take the mean across categories in the Google Mobility Reports. Different states may have different shares of categories. It would be good to also examine each category individually as robustness to make sure the results agree. Again, what is the correlation between the state-level estimates using the mean index and the state-level estimates using each individual category?

While we agree that in principle, separating out mobility for each category would be ideal, we aggregated to a mean in order to deal with missing data for many counties. For instance, only 44,000 county-day observations (out of 286,000) have non-missing data for visits to parks. Nonetheless, we examined each category individually and found that in general, the estimates across the categories are positively correlated with the exception of visits to the workplace. We hypothesize that this is because many essential workers continued to go to work.

In addition, we investigated whether variation in state-level estimates for these categories is associated with the various socioeconomic and epidemiological factors we identified for our primary analysis.

We have added the following to our statistical analysis section:

“Lastly, to understand how different categories of mobility (e.g., workplace, transit use etc.) responded to SIP orders, we re-ran our primary state-level RD specification with each mobility category as the outcome. We investigated variation in these coefficients across states, and investigated their association with the epidemiological and socioeconomic factors discussed earlier.”

We have added the following to our results section:

“We re-ran our primary RD specification using changes in mobility across each non-residential category separately and investigated the correlation of these measures. In general, all measures moved in the same direction as our index measure, and were generally strongly correlated with the index measure. The one exception was mobility at the workplace, which we hypothesize might be because essential workers continued to show up at work.”

As well as the following:

“As with our primary analysis, we find no relationship between cases or deaths for category-specific analyses. Consistent with the results for our index measure of mobility, Donald J. Trump vote share was associated with smaller reductions in mobility for parks (-20.8, 90% CI: -35.8 to -5.9) and transit (-3.45, 95% CI: -6.4 to -0.6). Our findings were similar for per capita income and poverty rate for certain categories as well. A unique finding from this analysis is that early SIP implementation appears to be associated with a large decline in mobility in the workplace category (7.54, 90% CI: 0.29 to 14.79).”

We have added the following paragraph to our discussion section:

“However, in examining category-specific mobility changes, we do find that earlier SIP implementation appears to be associated with larger declines in visits to workplaces. While we cannot identify the precise mechanism for this, it may be that some private responses may have been counter to the public response (e.g., employers attempting to stay open).”

And we have added tables to the appendix:

- The authors do not account for county-level SIPs. Different states may have different shares of county SIPs already---driving observed heterogeneity. See Allcott et al. (2020; "Economic and Health Impacts of Social Distancing Policies during the Coronavirus Pandemic") for treatment of county level SIPs.

While we agree with the reviewer that county-level SIP orders may have had some county-specific effects, this would not bias our results because once a state SIP is enacted, all counties become “treated.” County-level SIP orders certainly may have played a role in moderating the effectiveness of state-level SIP (likely reducing their effectiveness). However, because of the relatively small number of county-level SIP orders (a little over 300 in Allcott 2020) and the challenge in incorporating them correctly into an RD framework, we don’t believe that additional analyses is warranted.

We have added the following to our limitations section:

“Fourth, while some other work has relied on county-level SIP orders as well, our analysis has not accounted for these changes. However, because state-level SIP orders affect all counties within the state, county-level SIP orders are unlikely to bias our results. Rather, county-level SIP orders may act as a moderator possibly reducing the effectiveness of state-level SIP orders.”

Correlates of estimated treatment effects.

- Small sample size is not a good reason to change the standard of statistical significance. The authors should just report the coefficient estimates along with the standard errors. The reader can then easily perform any required statistical tests.

While we recognize that the appropriate reporting of confidence intervals and determination of statistical significance is a contentious issue, we disagree that small sample size is not an adequate reason to rely on a narrower confidence interval. Even R.A. Fisher (1925) suggested alpha levels of 0.10 may be appropriate for small samples. First, by using a narrower confidence interval, we reduce the margin of error of the conditional mean. Second, if we did use the full sample size of counties and weighted by population, it is likely that most of our bivariable regressions would (spuriously) produce statistically significant results.

- It would be useful to see multivariate analysis of all covariates in addition to the bivariate analysis that is already conducted.

Unfortunately, due sample size limitations (N=39), we do not have enough observations for the regression to run successfully with all covariates included on the right-hand side of the equation. 

The authors should really have standard RD plots for each state estimate in the appendix so the reader can visualize the extent to which they think the RD assumptions are valid.

We agree and have added a matrix of RD plots to the appendix.

Literature:

- Painter and Qiu (2020; "Political beliefs affect compliance with COVID-19 Social Distancing Orders") should be cited because it is most relevant to examining heterogenous effects in response to SIPs by partisanship.

- Please also go through https://scholar.google.com/scholar?cites=939500896822965987 to make sure all relevant literature is being cited for heterogenous effects of SIPs (e.g., Wright et al.).

- The findings in this manuscript should then be compared to what others have found.

We have added these citations and have added to our discussion some framing in context of what others have found.

“In addition, while not directly comparable, our analysis is consistent with recent work finding that income is associated with the ability to comply with social distancing orders. (12) Our results are also similar in magnitude to other work using event-study designs (22), and we come to similar conclusions that a large component of the overall mobility reduction occurred due to a private rather than policy-driven response.”

Comments from Reviewer 3

1. In Line 44, "SIP orders reduced mobility nationally by 12 percentage points."

There may have been text cut off; we are unsure of the reviewer’s specific concern but may incorporate any additional feedback in the next iteration.

2. The observations are Android cellphone users who have turned on location tracking. Are these observations representative of the population? Could the authors provide some summary statistics of demographics of such a sub-population?

Unfortunately, demographics of this subpopulation are not accessible to us. We have noted in the limitations section that the demographics of individuals represented by Google mobility data may not be representative of the overall population. In addition, we have referenced a report documenting some important demographic differences between Android and iPhone users.

We have added the following to the limitations section:

“Second, it is unclear whether the Android users who contribute data to Google Community Mobility Reports are representative of individuals living in that county. Unfortunately, this information is unavailable. Indeed, there are observed demographic differences between Android and iPhone users across age groups and gender.(24) To the extent they are different, our results may not generalize to other individuals.”

3. In Line 120, it is unclear how the index measure is constructed. The index measure is the mean of what?

We have clarified in line 115-118 that this is the mean of the percent change in mobility for all non-residential categories:

“We then computed an index measure by taking the mean of percent change for all non-residential categories which included: retail and recreation, groceries and pharmacies, parks, transit stations, and workplaces.”

4. In Line 129, the observations on the day of implementation are dropped. I wonder how the estimation results could change without dropping those observations. The SIP orders should be expected, so I am not sure if the washout period should be taken into account. The authors may give more justication or reference.

While we agree with the reviewer that in some cases, policy changes are anticipated, we do not believe this is the case with SIP orders. California, for instance, announced its SIP order on the same day that it was to be enacted (https://www.kqed.org/science/1959566/california-gov-gavin-newsom-orders-state-to-shelter-in-place). Massachusetts was in a similar situation (https://www.mass.gov/news/governor-charlie-baker-orders-all-non-essential-businesses-to-cease-in-person-operation). In these cases, a washout period accounts for potential short-term increase in mobility as people rush the stores (https://fortune.com/2020/04/20/coronavirus-retail-industry-ecommerce-online-shopping-brick-and-mortar-covid-19/), and also allows for a short time for people to respond to the order appropriately. We have added the following in the Study Variables section:

“In addition, when assessing other policies, there may be expectation of enactment. SIP orders, however, appear to have been announced with little to no delay between announcement and enactment.(9) Thus the announcement of these orders were associated with spikes in mobility as residents rushed to stock up on essential goods. (10) To account for this, and to capture actual response to the SIP order we allowed for a one-day washout period. We thus dropped observations on the day of implementation.”

5. In Figure 2, the label of the horizontal axis is \\Days from SIP Implementation." Shouldn't it be \\percentage change in mobility"? I don't see how the relative date of SIP implementation reflects in the figure. Is California the last to implement SIP, and is South Carolina the first?

We appreciate the opportunity to clarify. The reviewer is correct that the x-axis should say “Percentage Change in Mobility,” and we have modified it accordingly. The reviewer is also correct that the figure does not represent the relative date of implementation, and that this caption was meant to refer to Figure 1. We have modified both figures accordingly.

Comments from Reviewer 4

Major comments:

* An important factor that may explain the findings are differences in how the SIP and related measures were implemented in each state. For example, some states had more enforcement, more stringent measures such as restrictions on religious and other public events, maximum number of people allowed in social gatherings, closing of different types of businesses (indoor and outdoor dining, bars, gyms, stores), etc. Policies aiming at restricting mobility also differed widely at the county and city level in each state. Hence, part of the difference in effects may be due to differences in how the SIP and accompanying policies were implemented across states. The estimated differences across states may therefore represent responses to different policies (or bundles of policies), not differential responses to a homogeneous policy implemented uniformly across states.

We agree with the reviewer that details of policy implementation may have contributed to differences in effectiveness across states. This is related to a comment that Reviewer 1 made, noting that other policies were also enacted during this time. In an ideal scenario, we would have multiple components of the policy itself coded and be able to control for it in analysis. Because this is impractical, we investigated whether the state-level coefficients we estimated vary with indicators of other policies (school closures, daycare closures, nursing home restrictions, business closures, restaurant closures, and gym closures). Across these regressions, at the 90% confidence interval, only school and business closures were associated with variation in the RD estimate, both being associated with more effective SIP orders. 

We have added the following to the statistical analysis section:

“Because SIP orders were implemented alongside complementary policies in some states, we also assessed whether SIP impact differed by whether the state had implemented a range of other measures at the time of the SIP order.”

We have added this as a robustness check in the manuscript:

“We found some evidence that additional policies modified the effectiveness of SIP orders. School closures (-5.7%, 90% CI: -8.4 to -2.9) and non-essential business closures (-4.9%, 90% CI: -9.6 to -0.1) appear to be associated with more effective SIP orders. No other policies were significantly associated with variation in effectiveness of state-level SIP orders. (Table 2)” 

We have added the following to the discussion section:

“Less surprisingly, we found that certain policies – school closures and non-essential business closures – were complementary to SIP orders. It may be that these policies reduce burdens that would otherwise limit the ability of individuals to adhere with SIP orders.”

We have added the following to the limitations section:

“Third, other policies (such as transportation or business closures) were implemented either in tandem with or prior to SIP order enactment. These policies likely had their own effects, and may have affected compliance with SIP orders, which may explain some of the variation between states.”

* The differences between the estimates of mobility changes across states do not seem statistically significant in most cases (Figure 2). Hence much of this estimated variation may be due to sampling variability, not necessarily to real heterogeneity in the effect of the SIP orders.

While the reviewer is correct that there are a number of states where the effect of SIP orders was non-statistically significant, this is the case in one-quarter of states in this sample (10 out of 39). In addition, the results in our overall model is also driven by within-state variation.. 

* The regressions shown in Table 1 use only 39 observations, which seems very low. Lack of significance in the results for epidemiological factors may be due to lack of statistical power and not a true lack of effect. The statistical analysis does not convincingly justify the conclusions on page 12 (lines 265-273).

While we agree with the reviewer that small sample size may present an issue (and this is why we report 90% confidence intervals), we believe that our conclusion that post-SIP mobility changes were not simply a continuation of an existing trend are still valid. Indeed, we did find that certain moderating factors were significant, and those that weren’t significant had an estimated effect size closer to zero. This is broadly consistent with the visual inspection of the RD plot in Figure 1 and with our bivariable analysis. 

Minor comments:

* The data only include Android cellphone users. The authors should discuss whether they think this subgroup is representative of each state's population.

We agree that this is of possible concern. We have added a reference to an analysis of demographic differences between iPhone and Android users. We have also added the following discussion in the limitations section:

“Second, it is unclear whether the Android users who contribute data to Google Community Mobility Reports are representative of individuals living in that county. Unfortunately, this information is unavailable. Indeed, there are observed demographic differences between Android and iPhone users across age groups and gender.(24) To the extent they are different, our results may not generalize to other individuals.” 

* I was confused by the second paragraph on page 8 and the note under Table 1 stating that standard errors are clustered at the state level. My understanding is that the data for these regressions is already at the state level, which means each cluster has a single observation (so standard errors are heteroskedasticity-robust but do not effectively allow for clustering in any useful way).

The reviewer is correct, and we have corrected this typo.

* It would be interesting to analyze changes in mobility after the SIP is lifted to see whether mobility reverts back to its previous levels or remains lower.

We agree with the reviewer that this would be an interesting line of inquiry! Unfortunately, we leave this question to future researchers as an answer would require a substantial expansion of our existing analysis and is beyond the scope of our manuscript.

---

## [Decision Letter · Decision Letter 1]

16 Dec 2020

PONE-D-20-29575R1

Effectiveness of COVID-19 Shelter-in-Place Orders Varied By State

PLOS ONE

Dear Dr. Feyman,

Thank you for submitting your manuscript to PLOS ONE. After careful consideration, we feel that it has merit but does not fully meet PLOS ONE’s publication criteria as it currently stands. Therefore, we invite you to submit a revised version of the manuscript that addresses the points raised during the review process.

This is a difficult decision because 2 reviewers felt you addressed reviewers' concerns but 1 reviewer still recommended rejection. I don't usually go through multiple rounds of revision for PLoSOne but because the lone reviewer with ongoing critiques raised methodological concerns which impact the results and interpretation, I am returning the manuscript to you. Please provide a detailed rebuttal/revision in response to the reviewers' comments.

We look forward to receiving your revised manuscript.

Kind regards,

Jaymie Meliker, Ph.D.

Academic Editor

PLOS ONE

Reviewers' comments:

Reviewer's Responses to Questions

**Comments to the Author**

1. If the authors have adequately addressed your comments raised in a previous round of review and you feel that this manuscript is now acceptable for publication, you may indicate that here to bypass the “Comments to the Author” section, enter your conflict of interest statement in the “Confidential to Editor” section, and submit your "Accept" recommendation.

Reviewer #2: (No Response)

Reviewer #3: All comments have been addressed

Reviewer #4: (No Response)

2. Is the manuscript technically sound, and do the data support the conclusions?

Reviewer #2: Yes

Reviewer #3: Yes

Reviewer #4: No

3. Has the statistical analysis been performed appropriately and rigorously? 

Reviewer #2: Yes

Reviewer #3: Yes

Reviewer #4: No

4. Have the authors made all data underlying the findings in their manuscript fully available?

Reviewer #2: Yes

Reviewer #3: Yes

Reviewer #4: Yes

5. Is the manuscript presented in an intelligible fashion and written in standard English?

Reviewer #2: Yes

Reviewer #3: Yes

Reviewer #4: Yes

6. Review Comments to the Author

Reviewer #2: I found the authors' revision responsive to the majority of my comments. I just have two small comments:

1. I won't argue with the authors about the appropriate significance level and whether it should vary with sample size. I do, however, think a simple solution to people having different views on this would be to just add the standard errors to Tables 1 and 2 in addition to the 90% CI.

2. The point about county SIPs not biasing results depends on what the authors are trying to estimate. If the authors are trying to estimate the treatment effect of an entire state adopting a SIP relative to an entire state (including all counties) having no SIP, then pre-existing county SIPs are going to bias those estimates. I think the authors should drop claims regarding bias, and just note that pre-existing county SIPs may attenuate estimates.

Reviewer #3: The authors have addressed most of my comments and questions in the previous report.

1. Regrading the rst comment in my previous report: In Line 44 (now in Line 41),

"SIP orders reduced mobility nationally by 12 percentage points..." I think it should

be 12, rather than -12? Doesn't "reduce...by -12" mean \\increase...by 12"?

2. For the rst robustness check, running the RD using the covariates as the dependent

variable is called "validity check" in the literature, to my knowledge. This falsi-

cation test is for the identication assumptions to access whether the RD design is

valid. This is dierent from checking the robustness of the results against various

bandwidths.

Reviewer #4: The analysis of effect heterogeneity across states is still unconvincing. On the one hand, the authors estimate a large number of effects for different states, but do not provide any statistical test to show that the effects are significantly different between states. Here I do not mean testing whether the effects are significantly different from zero, which can easily be seen from the confidence intervals in Figure 2, but whether the effects are significantly different from each other, which is what the authors argue in their paper. In fact, most of the estimates in Figure 2 seem rather close to each other and their confidence intervals overlap in a large number of cases, which suggests that indeed many of these estimates may actually be capturing the same or very similar effects (perhaps with some exceptions such as Tennessee). Given that effect heterogeneity is the main point of this paper, the analysis of this issue seems overly superficial.

On the other hand, the regression analyses in Tables 1 and 2 also seem rather weak, primarily because they consist of many bivariate regressions. My understanding is that the authors cannot run multivariate regressions due to the very small sample size, but that is not a good justification of the bivariate analysis (rather, it could indicate that the available data are simply not rich enough to analyze the question of interest satisfactorily).

Moreover, even with such small sample sizes, the authors do find significant relationships between school and business closures and the size of the effect. This suggests that a large part of the between-state variability in the effects of the SIP may be simply capturing the fact that a different bundle of measures was implemented in each state. It could therefore be the case that, if the data were rich enough to account for these differences in policy implementation, the estimated heterogeneity in the effect of SIP could decrease or even disappear. I don't think this problem can be dismissed simply as a minor methodological limitation, as it is serious enough that it undermines the main point of the paper (namely, that the effect of the SIP orders on mobility varied by state).

7. PLOS authors have the option to publish the peer review history of their article (what does this mean?). If published, this will include your full peer review and any attached files.

Reviewer #2: No

Reviewer #3: No

Reviewer #4: No

---

## [Author Response · Author response to Decision Letter 1]

17 Dec 2020

Comments from Reviewer 2

1. I won't argue with the authors about the appropriate significance level and whether it should vary with sample size. I do, however, think a simple solution to people having different views on this would be to just add the standard errors to Tables 1 and 2 in addition to the 90% CI.

We agree with the reviewer that the appropriate significance level may be in the eye of the beholder. As suggested, we have added the standard errors to Tables 1 and 2.

2. The point about county SIPs not biasing results depends on what the authors are trying to estimate. If the authors are trying to estimate the treatment effect of an entire state adopting a SIP relative to an entire state (including all counties) having no SIP, then pre-existing county SIPs are going to bias those estimates. I think the authors should drop claims regarding bias, and just note that pre-existing county SIPs may attenuate estimates.

The reviewer’s point is well-taken. We have added the following to the limitations section:

“Any pre-existing county-level SIP orders may attenuate our effect estimates. Thus, our results may be a conservative estimate.”

Comments from Reviewer 3

1. In Line 44 (now in Line 41), "SIP orders reduced mobility nationally by 12 percentage points..." I think it should be 12, rather than -12? Doesn't "reduce...by -12" mean \\increase...by 12"?

We thank the reviewer for catching this typo! 

2. For the first robustness check, running the RD using the covariates as the dependent variable is called "validity check" in the literature, to my knowledge. This falsification test is for the identification assumptions to access whether the RD design is valid. This is different from checking the robustness of the results against various bandwidths.

We agree with the reviewer. We present both sets of analyses in our results and clarify their meaning:

In the results section we have written:

“While estimates from state-level models with a fixed bandwidth differed from estimates that allowed the bandwidth to vary by state, the estimated coefficients were strongly correlated (r=0.77)(Figure S2).”

As well as:

“To test whether disease burden is a confounder in our RD models, we examined whether it has a discontinuity with the running variable. Our results (0.48, 95% CI:-0.04 to 0.99) indicate that while it is plausible that SIP enactment is endogenous to disease burden, it is nonetheless not a confounder for our analysis.”

Comments from Reviewer 4

The analysis of effect heterogeneity across states is still unconvincing. On the one hand, the authors estimate a large number of effects for different states, but do not provide any statistical test to show that the effects are significantly different between states. Here I do not mean testing whether the effects are significantly different from zero, which can easily be seen from the confidence intervals in Figure 2, but whether the effects are significantly different from each other, which is what the authors argue in their paper. In fact, most of the estimates in Figure 2 seem rather close to each other and their confidence intervals overlap in a large number of cases, which suggests that indeed many of these estimates may actually be capturing the same or very similar effects (perhaps with some exceptions such as Tennessee). Given that effect heterogeneity is the main point of this paper, the analysis of this issue seems overly superficial.

The reviewer makes an important point; we had relied on the ‘eye test’ showing that many state confidence intervals do not overlap, and had not formally tested whether state-level estimates are statistically different from one another. For this revision, we conducted Z-tests comparing regression coefficients for every state pairing (see Clogg, C. C., Petkova, E., & Haritou, A. (1995). Statistical methods for comparing regression coefficients between models. American Journal of Sociology, 100(5), 1261–1293. https://doi.org/10.1086/230638). In the table below (now included in an Appendix), the results indicate that out of 741 comparisons, 429 (58%) are statistically different from one another.

The following has been added to the statistical analysis section (citations in manuscript):

“To investigate whether our state-level RD estimates are statistically different from one another, we conduct a series of z-tests for each state-pair in our sample.” 

We have also added the following to the results section:

“Among all state-pairs, our z-tests indicate 58% of regression coefficients were statistically different from one another (Table S6).”

On the other hand, the regression analyses in Tables 1 and 2 also seem rather weak, primarily because they consist of many bivariate regressions. My understanding is that the authors cannot run multivariate regressions due to the very small sample size, but that is not a good justification of the bivariate analysis (rather, it could indicate that the available data are simply not rich enough to analyze the question of interest satisfactorily).

The reviewer is correct that small sample sizes limit our ability to full investigate the extent of treatment effect heterogeneity, which we have acknowledged as a limitation in the revised manuscript:

 “Lastly, small sample sizes in our bivariable regressions may limit our ability to detect associations with state-level effect estimates. Despite this limitation, we are able to identify associations between our effect estimates and certain important measures such as political affiliation and some demographics. Moreover, the heterogeneity that we find is consistent with prior evidence on heterogeneity in mobility and responsiveness to SIP orders.”

Moreover, even with such small sample sizes, the authors do find significant relationships between school and business closures and the size of the effect. This suggests that a large part of the between-state variability in the effects of the SIP may be simply capturing the fact that a different bundle of measures was implemented in each state. It could therefore be the case that, if the data were rich enough to account for these differences in policy implementation, the estimated heterogeneity in the effect of SIP could decrease or even disappear. I don't think this problem can be dismissed simply as a minor methodological limitation, as it is serious enough that it undermines the main point of the paper (namely, that the effect of the SIP orders on mobility varied by state).

The reviewer is correct that SIP implementation varied across states. Unfortunately, a deeper dive into the differences across these 39 individual state SIP orders is outside the scope of this article.

However, we also believe that differences in concurrent policy implementation are unlikely to be the primary driver of this heterogeneity. Indeed, nearly all states in our sample (38 and 32, respectively) implemented the policies that have a relationship with our estimated coefficients (school closures and business closures). Moreover, even among states with every other policy implemented, there remains substantial variation in effect size (mean: -14.8, SD:11.5). We have additionally included the following in the discussion section:

“While the variation in the effectiveness of SIP orders in reducing mobility is likely to partly be a function of other policies implemented concurrently, this is unlikely to be the driver of the effect heterogeneity that we observe. Indeed, nearly all states in our sample (38 and 32, respectively) implemented the policies that have a relationship with our estimated RD coefficients (school closures and business closures). Moreover, even among states with every policy implemented, there remains substantial variation in effect size (mean: -14.8, SD:11.5) Future research should explore the potential mechanisms underlying these differences in more detail.”

---

## [Editor Report · Decision Letter 2]

21 Dec 2020

Effectiveness of COVID-19 Shelter-in-Place Orders Varied By State

PONE-D-20-29575R2

Dear Dr. Feyman,

We’re pleased to inform you that your manuscript has been judged scientifically suitable for publication and will be formally accepted for publication once it meets all outstanding technical requirements.

Kind regards,

Jaymie Meliker, Ph.D.

Academic Editor

PLOS ONE
---

## [Editor Report · Acceptance letter]

23 Dec 2020

PONE-D-20-29575R2 

Effectiveness of COVID-19 Shelter-in-Place Orders Varied By State 

Dear Dr. Feyman:

I'm pleased to inform you that your manuscript has been deemed suitable for publication in PLOS ONE. Congratulations! Your manuscript is now with our production department. 

Kind regards, 

on behalf of

Dr. Jaymie Meliker 

Academic Editor

PLOS ONE